# MACHINE TRUTH SERUM

## ABSTRACT

*Wisdom of the crowd* (Surowiecki, 2005) revealed a striking fact that the majority answer from a crowd is often more accurate than any individual expert. We observed the same story in machine learning - ensemble methods (Dietterich, 2000) leverage this idea to combine multiple learning algorithms to obtain better classification performance. Among many popular examples is the celebrated Random Forest (Ho, 1995), which applies the majority voting rule in aggregating different decision trees to make the final prediction. Nonetheless, these aggregation rules would fail when the majority is more likely to be wrong. In this paper, we extend the idea proposed in *Bayesian Truth Serum* (Prelec, 2004) that "a surprisingly more popular answer is more likely the true answer" to classification problems. The challenge for us is to define or detect when an answer should be considered as being "surprising". We present two machine learning aided methods which aim to reveal the truth when it is minority instead of majority who has the true answer. Our experiments over real-world datasets show that better classification performance can be obtained compared to always trusting the majority voting. Our proposed methods also outperform popular ensemble algorithms. Our approach can be generically applied as a subroutine in ensemble methods to replace majority voting rule.

## 1 INTRODUCTION

Wisdom of the crowd harnesses the power of aggregated opinion of a diverse group rather than a few individuals. Though initially proposed for mainly aggregating human judgements, this idea has been successfully implemented in the context of machine learning. In particular, ensemble learning was proposed and studied to improve prediction performance by combining several learning models to obtain better results compared to a single one (Dietterich, 2000). The developed ensemble techniques have shown consistent benefits in real-world machine learning applications, evidenced by the Netflix Competition (Bennett et al., 2007) and Kaggle competition. Popular ensemble methods include Boosting (e.g., AdaBoost (Freund & Schapire, 1997)), Bootstrap aggregating (bagging), Stacking (Bishop, 2006), and Random Forest (Ho, 1995).

The most popular, as well as simple, way to perform aggregation is via majority voting rule. The classical example is Random Forest, which outputs the majority answer from multiple trained decision trees. Inference methods (Raykar et al., 2010; Zhang et al., 2014; Liu et al., 2012; Zhou et al., 2012; 2014) have been applied to perform smarter aggregation that aims to outperform majority-voted answers. These methods often leverage homogeneous assumption of certain hidden models over a large number of data points in order to perform joint inference.

Nonetheless, all above methods rely on the assumption that the majority answer is more likely to be correct - this is also true for the more sophisticated inference models, as the inferences will mostly likely initiate based on majority-voted answers (when the algorithm has no prior information). While enjoying this assumption that majority tends to be correct, this claim is questionable in settings where special knowledge is needed to infer the truth, but it is owned by few individuals when they are not widely shared (Chen et al., 2004; Simmons et al., 2010; Prelec et al., 2017). Echoing to the above problem of aggregating human judgements, we face similar challenge when aggregating classifiers' predictions in machine learning. For example, we have a deep learning (Goodfellow et al., 2016) classification model which performs the best among multiple models when used in the ensemble method. For some data point, the classification result of this deep learning model may be the correct minority. In this situation, applying majority voting leads to wrong answers.

We aim to complement the literature via studying whether we can aggregate classifiers better than majority voting even when majority opinion is wrong. We also target a method that can operate over each data point separately without assuming homogeneous assumptions across a massive dataset.

The question sounds unlikely to resolve at a first look, but we are inspired by the seminal work *Bayesian Truth Serum* (BTS) (Prelec, 2004; Prelec et al., 2017) which approached this question in the setting of incentivizing and aggregating truthful human judgements. The core idea behind BTS is simple and elegant: the correctness of an answer does not rely on its popularity, but rather whether it is "surprisingly" popular or not - here an answer that has a higher posterior (computed from reports of the crowds) than its prior is taken as being "surprisingly" popular, and should be considered as the true answer. This argument has a very intuitive Bayesian reasoning: the signal that improves over its prior is more likely to be informative. Prelec et al. (2017) also argued that via eliciting a peer prediction information, which is defined as the fraction of "how many other people would agree with you" from each agent, he will be able to construct an informative prior to compare with the majority vote posterior aggregation. BTS operates over each single question separately, without seeing a large number of similar tasks (in order to leverage a certain homogeneity assumption).

In this paper, we make a connection between these two seemingly irrelevant topics, and extend the key idea in *Bayesian Truth Serum* to aggregating classifiers' predictions. The challenge is that we would not be able to elicit a belief from a classifier on "how many other classifiers would agree with themselves", which renders the task of computing the prior difficult. We proposed two machine learning aided algorithms to mimic the procedure of reporting the peer prediction information, which we jointly name as *Machine Truth Serum* (MTS). We firstly propose Heuristic Machine Truth Serum (HMTS). In HMTS, we pair each baseline classifier (an agent) with a regressor model, which is trained to predict the peer prediction information using a processed training dataset. With the predictions from the regressors, we will be able to apply the idea of BTS to decide on whether to adopt the minority as the answer via comparing the prior (computed using the regressor) and the posterior for each label. Then we proposed Discriminative Machine Truth Serum (DMTS). In DMTS, we directly train one classifier to predict whether adopting the minority as the answer or not. As for the training complexity of our algorithm, the training time of HMTS is linear in the number of label classes because of the training of extra regressors. DMTS will only need to train one additional classifier and both the training and the running time are almost the same as the basic majority voting algorithm. Therefore our proposed methods are very practical to implement and run.

Our contributions summarize as follows: (1) We propose Heuristic Machine Truth Serum (HMTS) and Discriminative Machine Truth Serum (DMTS) to complement ensemble methods, which can detect when minority should be considered the final prediction instead of the majority. (2) Our experiments over 6 binary and 6 multiclass classification real-world datasets reveal promising results of our approach in improving over majority voting. Our proposed methods also outperform popular ensemble algorithms. (3) To pair with our experimental results, we also provide analytical evidences for the correctness of our proposed approaches. (4) Our approaches can be generically applied in ensemble methods to replace simple majority voting rules.

The rest of the paper is organized as follows. Section 2 introduces some related works. Section 3 reviews preliminaries and BTS. Section 4 introduces our Machine Truth Serum approaches. Section 5 presents our experimental results. Section 6 concludes our paper.

## 2  RELATED WORK

Wisdom of the crowd (Surowiecki, 2005) are often considered as being more accurate than a few elite individuals in applications including decision making of public policy (Morgan, 2014), answering the questions on general world knowledge (Singh et al., 2002), and so on. Typical algorithms for extracting wisdom of the crowd are based on majority voting, and the assumption that the majority opinion is more likely to be correct (Surowiecki, 2005). There is another line of machine learning works on proposing inference methods, including Expectation Maximization method (Raykar et al., 2010; Zhang et al., 2014), Variational Inference (Liu et al., 2012; Chen et al., 2015), and Minimax Entropy Inference (Zhou et al., 2012; 2014) to crowdsourcing settings, aiming to uncover the true labels from the noisy labels provided by non-expert crowdsourcing workers. Most relevant to us, (Prelec, 2004; Prelec et al., 2017) proposed a Bayesian Truth Serum method to extract the subjective

judgment of minority expert by collecting not only people's judgements but also how many percentage of the population share the same opinion.

In machine learning, ensemble methods combining multiple learning algorithms usually performs better than any single method (Dietterich, 2000). Ensemble methods consist of a rich family of algorithms. For instance, AdaBoost (Freund & Schapire, 1997) and Random Forest (Ho, 1995) are two different and commonly used ones. AdaBoost tries to optimize weighted voting outcomes, while Random Forest train and test using the majority voting rule. But these popular ensemble methods will be wrong when the minority is the correct answer.

In both the setting of aggregating human judgements and classifiers' predictions, most works, except for (Prelec, 2004), would fail when the majority opinion is instead likely to be wrong. But BTS only works in the setting of aggregating human judgements by collecting subjective judgment data. Based on the ideas proposed by Prelec (2004) and Prelec et al. (2017), we proposed two machine learning aided algorithms to find the correct answer when it is minority instead of majority in the setting of classifiers' predictions. As our proposed methods are machine learning algorithms, they can be trained and the predictions will be made automatically instead of collecting subjective judgment data as the case in (Prelec, 2004).

## 3 PRELIMINARY

In this paper, we consider both binary and multiclass classification problems. Nonetheless, for simplicity of demonstration, our main presentation focuses on binary classification. A multi-class extension of our method is presented in Section 4.3.

Suppose that we have a training dataset $\mathcal{D} := \{(x_i, y_i)\}_{i=1}^N$ and a test dataset $\mathcal{T} := \{(x_i, y_i)\}_{i=1}^T$, where $x_i \in X \subseteq \mathbb{R}^d$ is a $d$-dimensional vector. We have $K$ baseline classifiers $\mathcal{F} := \{f_1, f_2, ..., f_K : X \to \{0, 1\}\}$ that map each feature vector to a binary classification outcome. Ensemble method such as boosting algorithms can combine $\{f_1, f_2, ..., f_K\}$ to get better prediction results than each single one. For instance, Random Forest first applies the bootstrap aggregating to train multiple different decision trees to correct overfitting problems of decision trees. After training, the majority rule will be applied to generate the prediction result.

The above dependence on the majority voting rule is ubiquitous in ensemble methods. The key assumption of using the majority rule is that the majority is more likely to be correct than random guessing. Denoting as $\mathsf{Maj}(\{f_1(x), f_2(x), ..., f_K(x)\})$ the majority answer from the $K$ classifiers, formally, most, if not all, methods require that $P(\mathsf{Maj}(\{f_1(x), f_2(x), ..., f_K(x)\}) \neq y) < 0.5$. Our goal is still to construct a single aggregator $\mathcal{A}(\{f_1, f_2, ..., f_K\})$ that takes the classifiers' predictions on each data point as inputs and generates an accurate aggregated prediction. But we aim to provide instruction to cases where it is possible that $P(\mathsf{Maj}(\{f_1(x), f_2(x), ..., f_K(x)\}) \neq y) > 0.5$. The challenge is to detect when the minority population has the true answer.

### 3.1 BAYESIAN TRUTH SERUM

(Prelec, 2004) considers the following human judgement elicitation problem: There are a set of agents denoted by $\{a_i\}_{i=1}^K$. The designer aims to collect subjective judgement from each agent about an unknown event $y \in \{0, 1\}$ and aggregate accordingly. Each of the agent $i$ needs to report his own predicted label $l_i \in \{0, 1\}$ for $y$, and the percentage of other agents he believes will agree with him $p_i \in [0, 1]$. We will also call this second belief information as the *peer prediction information*. Denote the belief of agent $i$ as $\mathcal{B}_i$. $p_i$ is defined as follows: $p_i = \mathbb{E}_{\mathcal{B}_i}\left(\frac{\sum_{j \neq i} \mathbb{1}(l_j = l_i)}{K-1}\right)$.

We, as the designer, obtain the prediction labels $\{l_i\}_{i=1}^K$ and the percentage information $\{p_i\}_{i=1}^K$ from all the agents. The posterior for each label is defined as the actual percentage of this label which can be easily calculated utilizing the prediction results: (for label 1)

$$\mathsf{Posterior}(1) = \frac{\sum_i \mathbb{1}(l_i = 1)}{K} \tag{1}$$

In (Prelec, 2004; Prelec et al., 2017), Prelec et al. promote the idea of using the average predicted percentage of the responding label as the approximation of the priors: (for label 1).

$$\mathsf{Prior}(1) = \frac{\sum_{i=1}^{K} p_i^{\mathbb{1}(l_i=1)} \cdot (1-p_i)^{1-\mathbb{1}(l_i=1)}}{K} \tag{2}$$

If $\mathsf{Posterior}(1) > \mathsf{Prior}(1)$, label 1 will be taken as the surprisingly more popular answer, which should be considered as the true answer $\hat{y}$, even though it might be in minority's hands. The same rule is applied to label 0. Formally, if we denote $\hat{y}$ as the aggregated answer:

$$\hat{y} = \begin{cases} 1 & \text{if } \mathsf{Prior}(1) < \mathsf{Posterior}(1); \\ 0 & \text{if } \mathsf{Prior}(1) > \mathsf{Posterior}(1). \end{cases} \tag{3}$$

The rest of the paper will focus on generalizing the above idea to aggregate classifiers' predictions.

## 4 MACHINE TRUTH SERUM

In this section, we introduce Machine Truth Serum (MTS). Suppose we have access to a set of baseline classifiers. Each classifier can be treated as an agent. We'd like to build a BTS-ish aggregation method to aggregate the classifiers' predictions. The challenge is to compute the priors from the classifiers - machine-trained classifiers do not encode beliefs as human agents do, so we cannot elicit the peer prediction information from them directly. We propose two machine learning aided approaches to perform the generation of this peer prediction information. We firstly introduce two MTS approaches for binary classification and then extend these approaches to multiclass classification case.

### 4.1 HEURISTIC MACHINE TRUTH SERUM

We first introduce heuristic machine truth serum (HMTS). The high level idea is to train a regression model for each classifier to predict the percent of the agreement from other classifiers on the prediction of each particular data point. After getting the predicted labels and the predicted peer prediction information of the classifiers, we can again approximate the priors using the predicted peer prediction information for each classifier, compute the average and compare it to posterior. In this part, HMTS for binary classification will be introduced firstly and its multiclass extension is stated in Section 4.3.

Given the training data $\mathcal{D} = \{(x_i, y_i)\}_{i=1}^{N}$ and multiple classifiers $\{f_j\}_{j=1}^{K}$, we first try to compute the $j$-th classifier's "belief" of the fraction of other classifiers that would "agree" with it. Denote this number as $\bar{y}_i^j$ for each training sample $(x_i, y_i)$. $\bar{y}_i^j$ can be computed as follows:

$$\bar{y}_i^j = \frac{\sum_{j \neq k} \mathbb{1}(f_j(x_i) = f_k(x_i))}{K-1}, \tag{4}$$

By above, we have pre-processed the training data to obtain $\mathcal{D}_j^H := \{(x_i, \bar{y}_i^j)\}_{i=1}^{N}$, $j = 1, ..., K$, which can serve as the training data to predict the peer prediction information of classifier $j$ (again to recall, peer prediction information is the fraction of other classifiers that classifier $j$ believes would agree with it). We then train peer prediction regression models $\{g_j\}_{j=1}^{K}$ on $\mathcal{D}_j^H := \{(x_i, \bar{y}_i^j)\}_{i=1}^{N}$, $j = 1, ..., K$ respectively to map $x_i$ to $\bar{y}_i^j$. We consider different class labels and will first train two regression models: $g_{j,0}$ and $g_{j,1}$ are two belief regression models of classifier $j$ and trained on the examples whose predicted labels are 0s ($\mathcal{D}_{j,0}^H := \{(x_i, \bar{y}_i^j) : f_j(x_i) = 0\}_{i=1}^{N}$) and 1s ($\mathcal{D}_{j,1}^H := \{(x_i, \bar{y}_i^j) : f_j(x_i) = 1\}_{i=1}^{N}$) respectively.

Then compute the following prior of label 1 for each $x_i$:

$$g_j(x_i) = \begin{cases} g_{j,1}(x_i) & \text{if } f_j(x_i) = 1; \\ 1 - g_{j,0}(x_i) & \text{if } f_j(x_i) = 0. \end{cases} \tag{5}$$

After obtaining these peer prediction regression models $g_j$s, the prior and posterior of $(x_i, y_i) \in \mathcal{T}$ in the test dataset are then calculated by,

$$\mathsf{Prior}(x_i, l = 1) := \frac{\sum_j g_j(x_i)}{K} \tag{6}$$

$$\mathsf{Posterior}(x_i, l = 1) := \frac{\sum_j \mathbb{1}(f_j(x_i) = 1)}{K} \tag{7}$$

---

**Algorithm 1** Heuristic Machine Truth Serum (Binary classification)

---

**Require:**
 1: Input:
 2: $\mathcal{D} = \{(x_1, y_1), ..., (x_N, y_N)\}$: training data
 3: $\mathcal{T} = \{(x_1, y_1), ..., (x_T, y_T)\}$: testing data
 4: $\mathcal{F} = \{f_1, ..., f_K\}$: classifiers
**Procedure:**
 5: Train $K$ classifiers ($\mathcal{F}$) on the training data
 6: **for** $j = 1$ to $K$ **do**
 7:     **for** $i = 1$ to $N$ **do**
 8:         Compute $\bar{y}_i^j$ according to Eqn.(4)
 9:     **end for**
10:     Train machine belief $g_{j,0}, g_{j,1}$ on training dataset $\mathcal{D}_j^H := \{(x_i, \bar{y}_i^j)\}_{i=1}^N$.
11: **end for**
12: **for** $t = 1$ to $T$ **do**
13:     Compute $\mathsf{Prior}(x_i, l = 1)$ and $\mathsf{Posterior}(x_i, l = 1)$ according to Eqn.(6) and Eqn.(7)
14:     **if** $\mathsf{Prior}(x_i, l = 1) < \mathsf{Posterior}(x_i, l = 1)$ **then**
15:         Output "surprising" answer 1 as the final prediction.
16:     **else if** $\mathsf{Prior}(x_i, l = 1) > \mathsf{Posterior}(x_i, l = 1)$ **then**
17:         Output "surprising" answer 0 as the final prediction.
18:     **end if**
19: **end for**

---

If $\mathsf{Prior}(x_i, l = 1) < \mathsf{Posterior}(x_i, l = 1)$, the "surprsing" answer 1 will be considered as the true answer. The decision rule is similar for label 0. The procedure is illustrated in Algorithm 1.

## 4.2 DISCRIMINATIVE MACHINE TRUTH SERUM

The Heuristic Machine Truth Serum above relies on training models to predict the peer prediction information for each classifier (which will be used to compute the priors) and compare them to the posteriors, and then decide on whether to follow the minority opinion or not. We notice the above task of determining whether to follow the minority or not is also a binary classification question. We can therefore utilize a classification model to directly predict for each data point whether the minority should be chosen as the answer or not.

We propose Discriminative Machine Truth Serum (DMTS). Again, DMTS for binary classification will be introduced firstly and its multiclass extension is stated in Section 4.3. With DMTS, a new training dataset $\mathcal{D}_D := \{x_i, \hat{y}_i\}_{i=1}^N$ about whether considering the minority as the final answer or not is constructed. Each data $\mathcal{D}_D := (x_i, \hat{y}_i)$, for $i = 1, ..., N$, in this new training dataset is calculated as follows: for each $(x_i, y_i) \in \mathcal{D}$

$$\hat{y}_i = \begin{cases} 1 & \text{if majority of } \mathcal{F} \text{ on } x_i \text{ is different from the true label;} \\ 0 & \text{if majority of } \mathcal{F} \text{ on } x_i \text{ is same as the true label.} \end{cases} \tag{8}$$

Now with above preparation, predicting whether majority is correct or not becomes a standard classification problem on $\mathcal{D}_D := \{x_i, \hat{y}_i\}_{i=1}^N$. This is readily solvable by applying standard techniques. In our experiments, we will mainly use a Multi-Layer Perceptron (MLP) (Goodfellow et al., 2016) denoted as $\bar{f}$. $\bar{f}$ is trained on this new training dataset and can directly predict whether we should adopt the minority as the answer or not. $\bar{f}$ does not restrict to MLP and can be other classifiers. We have tried several other methods, such as logistic regression, and similar conclusions are obtained. The procedure is further illustrated in Algorithm 2 in Appendix A.3.

## 4.3 MULTICLASS EXTENSION OF HMTS AND DMTS

HMTS and DMTS can be extended to multiclass classification problem with the same ideas by modifying them accordingly. In the multiclass case, $l \in \mathcal{C} = \{0, 1, ..., L\}$ is denoted as the class label of the dataset. Consider HMTS first. For each classifier $j$, we need to consider different class labels of regression models $\{g_{j,l}\}$, where $l \in \mathcal{C} = \{0, 1, ..., L\}$. $g_{j,l}$ is the belief regression model of classifier $j$ and trained on the examples whose predicting labels are $l$s.

Again compute the following prior for each $x_i$

$$g_{j,l}^*(x_i) = \begin{cases} g_{j,l}(x_i) & \text{if } f_j(x_i) = l; \\ \left(1 - g_{j,f_j(x_i)}(x_i)\right) \cdot ratio_l & \text{if } f_j(x_i) \neq l, \end{cases} \tag{9}$$

where $ratio_l = \frac{g_{j,l}(x_i)}{\sum_{c \in \mathcal{C}: c \neq f_j(x_i)} g_{j,c}(x_i)}$ is defined as the ratio of the $l$'s belief to the summation of all the other classes' beliefs except for the predicted class utilizing majority rule.

In HMTS, Eqn. (6) and (7) can be modified to the following:

$$\mathsf{Prior}(x_i, l = c) := \frac{\sum_{j=1}^{K} g_{j,c}^*(x_i)}{K} \tag{10}$$

$$\mathsf{Posterior}(x_i, l = c) := \frac{\sum_{j=1}^{K} \mathbb{1}(f_j(x_i) = c)}{K} \tag{11}$$

We then compute all the priors and posteriors of each class label based on Eqn. (10) and (11). It is possible that there exist more than one class labels whose posterior is larger than its prior. We define the set containing all these label classes as

$$\mathcal{C}_{sat} = \{c \mid \mathsf{Prior}(x_i, l = c) < \mathsf{Posterior}(x_i, l = c), c \in \mathcal{C}\}.$$

We predict the class label which has the biggest improvement from its prior to posterior:

$$\mathrm{argmax}_{c \in \mathcal{C}_{sat}} \left| \mathsf{Posterior}(x_i, l = c) - \mathsf{Prior}(x_i, l = c) \right|.$$

In DMTS, firstly we need to train a model that decides whether to apply the minority as the final answer which are very similar to the binary case. The difference is that we will then choose the minority answer as the predicted answer instead of using majority if i) it has the most votes in the minority answers and ii) the prediction result of classifier obtained in the training phase is 1 (we should use minority).

### 4.4 THEORETICAL ANALYSIS

We conduct a formal analysis about the correctness of our proposed algorithms. Not surprisingly, the key ideas of the proofs are adapted from the proof for BTS (Prelec et al., 2017). For simplicity, we only present the theorems for binary classification. The proofs of multiclass ones are similar to the binary case. The details of proofs are left to the Appendix A.1.

To set up for presenting the theorems, we restate our problem: we assume that each $x_i$ can take on any value in the discrete set $\{s_1, ..., s_m\}$ for the simplicity of proof. In practice, conceptually each feature vector can be represented by an assigned (large-enough) categorical number. One can consider $s_k(k = 1, 2, , , m)$ as a code for each feature vector. The proof based on continuous value can be deduced similarly.

Here we have two worlds $w_{io}$ (o = 0 or 1) of different class labels for any $x_i$. One world is actual, the other one is counterfactual. If we say $w_{i1}$ is the actual world for $x_i$, it means the predicting answer of $x_i$ in this world is 1 and $y = 1$ is also the ground truth label of $x_i$. $w_{i0}$ is the counterfactual world and the predicting answer of $x_i$ in this world is 0. In this paper, we are considering infinite samples. While finite samples is practical setting, it is important to first analyze and conclude some deductions in the infinite sample ideal case.

**Theorem 4.1.** *No algorithm exists for deducting the correct classification answer relying exclusively on feature vector distribution of true class label, $P(s_k|y_{o^*}), k = 1, ..., m$ and correctly computed posterior distribution over all possible classification labels given feature vectors, $P(y_o|s_k), k = 1, ..., m, o = 0, 1$ for any $x_i$. $o^*$ is the true class label.*

**Theorem 4.2.** *For any $x_i$, the average estimate of the prior prediction for the correct classification answer will be underestimated if not every classifier provides the correct classification prediction. Therefore, the minority should be the final answer instead of the majority if the prior (estimated prediction) is less than the posterior.*

Theorem 1 indicates that exclusively posterior probabilities based methods such as majority voting can not infer the true answer for all the time. In Theorem 2, the posterior and prior are the prediction distribution of other classifiers for each classifier - both are provided by our proposed MTS algorithms.

**Complexity**  For HMTS, for example in our experiments, another $15 \cdot (L + 1)$ (label classes $\{0, 1, ..., L\}$) simple regressors will be trained to predict others' beliefs based on 15 baseline classifiers. So the total training time is linear in the number of label classes. After training the extra regressors, running the algorithm only requires taking $L + 1$ averages (15 of the $15 \cdot (L + 1)$ regressors each) and compare with average posterior. DMTS will only need to train one additional classifier based on 15 classifiers and both the training and the running time are almost the same as the basic majority voting algorithm. The above complexity analysis shows our methods are very practical.

## 5 EXPERIMENTS

In this section, we present our experimental results. Particularly we test our proposed two MTS algorithms on 6 binary and 6 multiclass real-world classification datasets. Experimental results show that consistently better classification accuracy can be obtained compared to always trusting the majority voting outcomes.

### 5.1 DATASETS

In this section, 6 binary and 6 multiclass classification benchmark datasets (Pang & Lee, 2004) are used to conduct the experiments. The statistical information of these datasets are described in Table 4 in Appendix A.2. In this paper, each of the datasets we used has a small size - we chose to focus on the small data regime where the classifiers are likely to make mistakes. This is a better fit to our setting where majority opinion can be wrong with a good chance. For the splitting of training and testing, we used the original setting for the datasets providing training and testing files separately. For other datasets, only one data file is given. For the testing results' statistical significance, more data is distributed to testing dataset and 50/50 is considered as the splitting of training and testing.

### 5.2 EXPERIMENTAL SETUP AND RESULTS

Table 1: Overall accuracy and the number of increased correct predictions in the "high disagreement" cases of Uniformly-weighted Majority Voting, HMTS, and DMTS on the 6 binary classification datasets. The "high disagreement" means that the difference between the number of predicting 0 and 1 is small. We have 15 classifiers and the instance will be considered as having "high disagreement" if the vote number of majority class is 8 or 9. In HMTS, the numbers of "high disagreement" instances we consider in 6 datasets are **51, 52, 37, 149, 43, 45**. As for DMTS, the numbers are **30, 64, 37, 76, 13, 38**. The results having the highest accuracy for each dataset are highlighted.

| Datasets | Breast cancer | Hill Valley | Movie Review | Spambase | Australian | German |
|---|---|---|---|---|---|---|
| Majority | 92.96% | 80.86% | 80.10% | 73.57% | 81.74% | 76.00% |
| HMTS | **96.13%** (+8/51) | **81.69%** (+5/52) | **80.85%** (+5/37) | 76.87% (+48/149) | **83.44%** (+6/43) | **77.20%** (+6/45) |
| DMTS | 94.01% (+4/30) | 81.03% (+1/64) | 80.60% (+5/37) | **77.35%** (+70/76) | 82.94% (+4/13) | 76.20% (+1/38) |

In our binary classification experiments, we consider 5 commonly used binary classification algorithms which are Perceptron (Rosenblatt, 1958), Logistic Regression (LR) (Peng et al., 2002), Random Forest (RF), Support Vector Machine (SVM) (Chang & Lin, 2011), and MLP. In order to test the usefulness of our methods, we experiment with a noisy environment - we flipped the true class label with three noisy rates to construct three binary classifiers for each of the 5 methods which have mediocre performance on the test datasets. We wanted to diversify our classifiers by introducing different noisy rates (varying the data distribution). Our experiments used 0.06, 0.08, and 0.1 (probability of flipping the label) for each family of classifier. We also tried other values such as 0.1, 0.2, and 0.3, and we reached similar conclusions. In total, 15 different classifiers are obtained as the baseline classifiers.

The experimental results on the 6 binary classification datasets are reported in Table 1. From these results, we observe that Heuristics Machine Truth Serum (HMTS) tends to have more robust and better performances than Discriminative Machine Truth Serum (DMTS) in most datasets, especially in the small-size datasets. These can be explained by the fact DMTS itself is a MLP classifier which

needs a larger size of data to get good results. That HMTS can improve the classification accuracy in the small size of dataset is particularly useful in some fields such as healthcare in which collecting data is very time-consuming and expensive. As for the running time, DMTS is faster than HMTS as HMTS needs to compute the peer prediction results of all the 15 classifiers and DMTS only predicts once.

Table 2: Overall accuracy and the number of increased correct predictions in the "high disagreement" cases of Uniformly-weighted Majority voting, HMTS, and DMTS on the 6 multi-class classification datasets. We have 15 classifiers and the instance will be considered as having "high disagreement" if the vote number of majority class is less or equals to 6 for the 3-class and 4-class datasets. The threshold number is 5 for 6-class and 3 for 10-class datasets. In HMTS, the numbers of "high disagreement" instances we consider in 6 datasets are **61, 54, 27, 65, 15, 157**. As for DMTS, the numbers are **48, 45, 25, 81, 25, 40**. The results having the highest accuracy for each dataset are highlighted.

| Datasets | Abalone | Waveform | Wall-Following | Statlog | Optical | Pen-Based |
|---|---|---|---|---|---|---|
| # of class | 3 | 3 | 4 | 6 | 10 | 10 |
| Majority | 51.25% | 85.04% | 90.22% | 86.70% | 97.50% | 95.08% |
| HMTS | **51.45%** (+4/61) | 85.48% (+11/54) | 90.26% (+1/27) | **87.10%** (+8/65) | 97.61% (+2/15) | **95.57%** (+17/157) |
| DMTS | 51.40% (+3/48) | **85.60%** (+14/45) | **90.59%** (+10/25) | 86.75% (+1/81) | **97.66%** (+3/25) | 95.51% (+15/40) |

We also tested our extension to multi-class classification problems. Experimental results on 6 multi-class classification datasets are reported in Table 2. We observe that HMTS and DMTS obtained similarly good performance in the accuracy metric because the size of multi-class classification datasets is larger and the MLP of DMTS can perform better than the binary case.

Table 3: Comparison between popular ensemble and our proposed approaches

| Methods | Adaboost | Random Forest | Weighted Majority | Stacking | HMTS | DMTS |
|---|---|---|---|---|---|---|
| Breast Cancer | 94.37% | 94.37% | 94.01% | 94.72% | **96.13%** | 94.01% |
| Hill Valley | 62.05% | 57.93% | 53.30% | 62.38% | **81.69%** | 81.03% |
| Movie Review | 75.10% | 77.20% | **81.60%** | 70.30% | 80.85% | 80.60% |
| Spambase | 74.74% | 74.65% | 74.17% | 75.91% | 76.87% | **77.35%** |
| Australian | 82.03% | 84.06% | 84.06% | **85.22%** | 83.44% | 82.94% |
| German | 72.20% | 74.80% | 73.80% | 77.20% | **77.20%** | 76.20% |
| Abalone | 48.70% | 54.25% | **55.35%** | 54.55% | 51.45% | 51.40% |
| Waveform | 81.80% | 82.60% | 85.36% | 84.00% | 85.48% | **85.60%** |
| Wall-Following | **99.00%** | 95.22% | 95.37% | 89.07% | 90.26% | 90.59% |
| Statlog | 85.85% | 86.15% | 86.85% | 82.70% | **87.10%** | 86.75% |
| Optical | 93.99% | 94.88% | 92.21% | 95.83% | 97.61% | **97.66%** |
| Pen-Based | 94.97% | 95.45% | 90.59% | 95.43% | **95.57%** | 95.51% |

Finally, we compare between several popular ensemble algorithms and our proposed approaches. We list the testing accuracy for Adaboost with 15 decision tree base estimators, Random Forest with 15 decision trees, Weighted Majority(Germain et al., 2015), Stacking with the same setting of 15 classifiers utilized in our two MTS algorithms and Logistic Regression or SVM as meta classifier, HMTS, and DMTS for all 12 datasets in Table 3. As shown in the table, HMTS and DMTS outperform Adaboost, Random Forest, Weighted Majority, and Stacking in 8 datasets and are very close to the best in 3 datasets. An outlier dataset is Wall-Following and we found that decision tree based methods can get perfect performance on it. Compared to other weighted methods, we'd like to note that our aggregation operates on each single task separately - this means that our method will be more robust when the difficulty levels of tasks differ drastically in the dataset. None of the other weighted methods (with fixed and learned weights) has this feature. We also find that our method is robust to a smaller number of classifiers, in contrast to, say Adaboosting.

## 6 DISCUSSION AND CONCLUDING REMARKS

This paper proposes two machine learning aided methods HMTS and DMTS to detect when the minority should be the final answer instead of majority. Our experiments over 6 binary and 6 multiclass real-world datasets show that better classification performance can be obtained compared to always trusting the majority voting. Our proposed methods also outperform popular ensemble algorithms on three randomly selected datasets and can be generically applied as a subroutine in ensemble methods to replace majority voting. For future work, we plan to try more types of classifiers, especially the recent deep learning models, to train the belief models for baseline classifiers and apply our methods to more real-world datasets.

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

## A  APPENDIX

### A.1  PROOF OF THEOREMS IN SECTION 4.4

In this part, we provided the detailed proof of two theorems which are the analytical evidences for the correctness of our proposed approaches. For simplicity, we only show the proof details of binary classification. The proof of multiclass classification is similar to the binary case. This proof is largely adapted from (Prelec et al., 2017). Nonetheless we reproduce the details for completeness.

**Theorem A.1.** *No algorithm exists for inferring the correct classification answer relying exclusively on feature vector distribution of true class label, $P(s_k|y_{o^*}), k = 1, ..., m$ and correctly computed posterior distribution over all possible classification labels given feature vectors, $P(y_o|s_k), k = 1, ..., m, o = 0, 1$ for any $x_i$. $o^*$ is the true class label.*

*Proof.* In this proof, for any arbitrarily selected class label, we can construct a world model in which this selected class label is predicted as the answer and it is also the ground truth class label. And this world model can also generate feature vector distribution of true class label and correctly computed posterior distribution over all possible classification labels given feature vectors.

Based on the description of theorem, $P(s_k|y_{o^*}), k = 1, ..., m$ and $P(y_o|s_k), k = 1, ..., m, o = 0, 1$ are known. But we don't know which class label is the correct answer $y_{o^*}$. We can arbitrarily selected any class label $y_o$ as the ground truth class label. In the following part, a corresponding world model $Q(s_k, y_o)$ which can generate the known $P(s_k|y_{o^*})$ and $P(y_o|s_k)$ will be constructed.

Because the known parts don't constrain the prior over the feature vector - these priors can model differences in the baseline classifiers. In particular, we can set the prior to:

$$Q(s_k) = \frac{P(s_k|y_{o^*})}{P(y_o|s_k)} \left( \sum_r \frac{P(s_r|y_{o^*})}{P(y_o|s_r)} \right)^{-1}, k = 1, ..., m$$

Because posteriors in the constructed world model must equal to known posteriors: $Q(y_o|s_k) = P(y_o|s_k)$, for $k = 1, ..., m, o = 0, 1$. So we can get the joint distribution of answer $y_o$ and the feature vector $s_k$ in the constructed world model:

$$Q(y_o, s_k) = Q(y_o|s_k)Q(s_k) = P(s_k|y_{o^*}) \left( \sum_r \frac{P(s_r|y_{o^*})}{P(y_o|s_r)} \right)^{-1}$$

Then we can get the marginal distribution $y_o$ in the constructed world by summing over k:

$$Q(y_o) = \sum_k P(s_k|y_{o^*}) \left( \sum_r \frac{P(s_r|y_{o^*})}{P(y_o|s_r)} \right)^{-1} = \left( \sum_r \frac{P(s_r|y_{o^*})}{P(y_o|s_r)} \right)^{-1}$$

After getting the marginal distributions $Q(s_k), Q(y_o)$, and the matching posteriors, $Q(y_o|s_k) = P(y_o|s_k)$, for $k = 1, ..., m$, the feature vector distribution of true class label in the constructed world, $Q(s_k|y_o)$ can be calculated by:

$$Q(s_k|y_o) = \frac{Q(y_o|s_k)Q(s_k)}{Q(y_o)} = P(s_k|y_{o^*})$$

Because $y_o$ was arbitrarily chosen, this theorem is proved. $\square$

Theorem 1 shows that any algorithm relying exclusively on feature vector distribution of true class label and correctly computed posterior distribution over all possible classification labels given feature vectors (e.g. majority voting) can not deduct the correct classification answer.

In the following part, we are considering the extra information which is the estimation of other classifiers' prediction results. We use $P(v_o|s_k)$ to represent the how many percentage of classifiers will predict $y_o$ given $s_k$. We also define world classification function $W(s_k) = P(w_o|s_k)$. Two thresholds $c_0$ and $c_1 = 1 - c_0$ are given to make the final classification result. The classification rule is as follows:

$$W(s_k) = \begin{cases} w_0 & \text{if } P(w_0|s_k) > c_0; \\ w_1 & \text{if } P(w_1|s_k) > c_1. \end{cases}$$

**Theorem A.2.** *For any $x_i$, the average estimate of the prior prediction for the correct classification answer will be underestimated if not every classifier provides the correct classification prediction .*

*Proof.* We first prove that the actual percentage of correctly predicted classifiers for the true answer in the actual world exceeds counterfactual world's percentage for the true answer, $P(v_{o^*}|w_{o^*}) > P(v_{o^*}|w_k), k \neq o^*$.

By the definition of $W(s_k)$, we can get $P(w_{o^*}|v_{o^*}) > c_{o^*}, P(w_{o^*}|v_k) < c_{o^*}$. Then we have $P(w_{o^*}|v_{o^*})P(v_k) > P(w_{o^*}|v_k)P(v_k)$. So

$$P(w_{o^*}|v_{o^*}) > P(w_{o^*}|v_{o^*})P(v_{o^*}) + P(w_{o^*}|v_k)P(v_k) = P(w_{o^*}) \tag{12}$$

According to Bayesian rule, we have the following deduction:

$$\frac{P(v_{o^*}|w_{o^*})}{P(v_{o^*}|w_k)} = \frac{P(w_{o^*}|v_{o^*})P(w_k)}{P(w_k|v_{o^*})P(w_{o^*})} = \frac{P(w_{o^*}|v_{o^*})}{1 - P(w_{o^*}|v_{o^*})} \frac{1 - P(w_{o^*})}{P(w_{o^*})} \tag{13}$$

Based on (12), (13) is greater than one. So $P(v_{o^*}|w_{o^*}) > P(v_{o^*}|w_k), k \neq o^*$ is proved.

The estimate of classification prediction given the feature value $s_j$ can be computed by marginalizing the actual and counterfactual worlds, $P(v_{o^*}|s_j) = P(v_{o^*}|w_{o^*})P(w_{o^*}|s_j) + P(v_{o^*}|w_k)P(w_k|s_j)$. And we proved that $P(v_{o^*}|w_{o^*}) > P(v_{o^*}|w_k), k \neq o^*$. Therefore, $P(v_{o^*}|s_j) \leq P(v_{o^*}|w_{o^*})$. It will be the strict inequality unless $P(w_{o^*}|s_j) = 1$. Because some feature vectors will lead to strict inequality, the average estimate of the prior prediction will be strictly underestimated. This theorem is proved.

$\square$

## A.2 DATASETS

In this section, the detailed information of 6 binary and 6 multiclass classification datasets are described in Table 4. '#' of Inst. stands for the number of instances. '#' of Attr. stands for the number of attributes. '%' of Maj stands for the percentage of majority class.

## A.3 ALGORITHM 2 IN SECTION 4.2

In this part, we provided the detailed algorithm description for DMTS in Algorithm 2.

Table 4: Statistics of 6 binary and 6 multiclass classification datasets

| Data set | Breast cancer | Movie Review | German | Australian | Hill Valley | Spambase |
|---|---|---|---|---|---|---|
| # of Inst. | 569 | 1000 | 1000 | 690 | 606 | 4601 |
| # of Attr. | 30 | 77 | 24 | 14 | 100 | 57 |
| % of Maj. | 62.7% | 50% | 70.0% | 67.8% | 50.7% | 60.6% |

| Data set | Abalone | Waveform | Wall-Following | Stalog landsat | Optimal | Pen-Based |
|---|---|---|---|---|---|---|
| # of Inst. | 4177 | 5000 | 5456 | 4435 | 3823 | 7494 |
| # of Attr. | 8 | 21 | 24 | 36 | 62 | 16 |
| % of Maj. | 34.6% | 33.9% | 40.4% | 24.2% | 10.2% | 10.4% |

---

**Algorithm 2** Discriminative Machine Truth Serum (Binary classification)

---

**Require:**
 1: Input:
 2: $\mathcal{D} = \{(x_1, y_1), ..., (x_N, y_N)\}$: training data
 3: $\mathcal{T} = \{(x_1, y_1), ..., (x_T, y_T)\}$: testing data
**Procedure:**
 4: **for** $i = 1$ to $N$ **do**
 5:     Compute $\hat{y}_i$ according to Eqn.(8)
 6: **end for**
 7: Train DMTS classifier $\overline{f}$ on the dataset $\{x_i, \hat{y}_i\}_{i=1}^{N}$
 8: **for** $t = 1$ to $T$ **do**
 9:     Compute the classification result $\overline{y}_t := \overline{f}(x_t)$
10:     **if** $\overline{y}_t = 0$ **then**
11:         Stay with the majority answer.
12:     **else if** $\overline{y}_t = 1$ **then**
13:         Predict with the minority answer.
14:     **end if**
15: **end for**

---

