# OpenReview forum: "Machine Truth Serum"
_ICLR.cc/2020/Conference — Reject_

### Official Review · AnonReviewer1 · 2019-10-22
**Official Blind Review #1**

**Rating:** 1

**Review:**

Summary: This paper proposes two machine learning adaptations of the Bayesian truth serum approach to aggregating predictions from human experts. The first method proposed involves training two regression models for each classifier in the ensemble that predicts the proportion of other classifiers that assign the same label to a novel instance. The second approach is to train a binary classifier that, based on the features associated with an instance, determines whether the most common or second most common prediction made by individual ensemble members should be the prediction made by the ensemble.

Pros:
+ The core idea of adapting Bayesian truth serum to ensemble prediction in machine learning seems sensible
+ There is some evidence that the methods have an advantage over other common ensemble approaches in practice
+ Although there are quite a few small English mistakes, the paper is well structured and generally quite easy to follow

Cons & questions:
1. The theorem statements and proofs are underwhelming. First they seem quite vague. Second, it’s not clearly spelled out how the theorems relate to the presented method, and whether they really says anything useful about its correctness or efficacy. At face value they do not obviously analyse the correctness of the algorithm as claimed on pg6.
2. The paper contains some simple experiments, but I do not believe they are an adequate enough evaluation of the proposed approaches. The most useful comparison are the results given in Table 3, but only three datasets are used, the margins are small, error bars are not provided, and no significance testing is performed.
2.1 Standard practice when comparing multiple classifiers on multiple datasets would be to employ Friedman/Nemenyi post-hoc tests to determine relative performance of methods---see "Statistical Comparisons of Classifiers over Multiple Data Sets" by Janez Demšar (JMLR, 2006).
2.2 At minimum we expect Tab 3 to report results for all datasets used in the earlier experiments.
2.3 The experiments compare with AdaBoost, Random Forest, and Weighted Majority. I feel that stacking is probably the most interesting baseline to compare with, as this is a method for learning how to aggregate predictions from ensemble members.

Other:
A. This paper does not directly deal with representation learning, so is only loosely relevant to ICLR.
B. Not clearly unpacked why regression models need to be trained to predict \hat{y}_i^j (Equation 4) when this quantity can be computed at test time without knowing the ground truth label?
C. I do not understand the "X out of Y" description given in the caption to Table 1, which makes the results in this table hard to interpret. What is meant by "classifiers' disagreement is high enough"?


**Experience Assessment:**

I have published one or two papers in this area.

**Review Assessment: Checking Correctness Of Derivations And Theory:**

I assessed the sensibility of the derivations and theory.

**Review Assessment: Checking Correctness Of Experiments:**

I assessed the sensibility of the experiments.

**Review Assessment: Thoroughness In Paper Reading:**

I read the paper at least twice and used my best judgement in assessing the paper.

---

> ### Author Response · Authors · 2019-11-09
> **Response to Official Blind Review #1**
>
> Thank you very much for the valuable feedback.
>
> Question 1: Theorem 2 indicates we can know that the answer should be true if its prior is less than posterior through calculating extra prior information to get prior and conducting the subsequent comparison between prior and posterior. Based on these two theorems mentioned in the paper, the correctness of our proposed HMTS and DMTS are guaranteed theoretically. We have refined the statement.
>
> Question 2:
>
> - We showed the performance of randomly selected 3 datasets due to the space limitation. The experimental results in all 12 datasets will be added in the revision version.
> - We have added comparison with the stacking method, and have observed consistent improvement.
>
> We'd like to emphasize that though the overall improvements seem to be incremental, we want to emphasize that this is mainly due to the relatively small number of instances the majority of classifiers got it wrong. For instance, suppose there are 20% of data points that the majority classifiers got them wrong. If our algorithms achieve consistently 10% improvement, it will be a 2% overall improvement. Our method is primarily designed to improve the robustness of ensemble methods against these “hard” or challenging instances, which are arguably the more interesting cases.
>
> Others:
> B (“Not clearly unpacked why regression models need to be trained to predict \hat{y}_i^j (Equation 4) …...”): Counting over test data gives you a prior that is the same as posteriors because it's not actual estimated priors. It is very easy to verify using simple algebra.
>
> C. (“I do not understand the "X out of Y" description given in the caption to Table 1, which makes the results in…...”): ''X'' in the ''X out of Y'' means the number of increased correct predictions by applying our new algorithms compared to using majority voting. 'Y'' is the number of instances at which the classifiers' disagreement is high enough. Then we will explain what "classifiers' disagreement is high enough" means. For example, we have 15 basic classifiers for a binary-class classification problem. "8:7" means 8 classifiers predicted 1 and 7 predicted 0. "8:7" has more disagreements than, for example, "14:1". If the difference between the number of predicting 1 and predicting 0 is small we call it "classifiers' disagreement is high enough". In binary classification experiments, ''8:7'', ''7:8'', ''9:6'', and ''6:9'' are considered as ''high disagreement''.
>
> We have now better presented Table 1 & 2 (we believe).

---

### Official Review · AnonReviewer3 · 2019-10-24
**Official Blind Review #3**

**Rating:** 3

**Review:**

Inspired by work in ensembling human decisions, the authors propose an ensembling technique called "Machine Truth Serum" (based off "Bayesian Truth Serum"). Instead of using majority vote to ensemble the decisions of several classifiers, this paper follows the "surprisingly popular" algorithm; the ensembled decision is the decision whose posterior probability (based on several classifiers) most exceeds a prior probability (given by classifier(s) trained to predict the posterior predictions). It's quite a nice idea to bring this finding from human decision-making to machine learning. If it worked in machine learning, it would be quite surprising, as the surprisingly popular algorithm risks that the ensemble makes a decision against the majority vote, which is usually consider the safe/default option for ensembling.

Overall, I did not find the experiments (in the current state) to provide compelling enough support for the claim that MTS is a useful approach to ensembling in machine learning.
* Unless I am mistaken, the authors use a more powerful model (an MLP) as the regressor compared to some of the models they ensemble over. In practice, people ensemble the most powerful models they have available, so it's unclear if using a regressor with the same capacity as the ensembled classifiers will provide any additional benefit. On a related note, it would be nice to know what is the classification performance of each individual classifier? As well as how often the regressor correctly choose to go with several weaker models rather than the strongest model. In particular, I am concerned that the performance of the ensemble might be less than or equal to the performance of a single MLP classifier (or whatever other model does best).
* "In this paper, each of the datasets we used has a small size - we chose to focus on the small data regime where the classifiers are likely to make mistakes." Why not try large data tasks that are challenging for state-of-the-art models? The paper makes a general claim that MTS is a good way to aggregate predictions, so only evaluating on small datasets seems to be a limitation
* As I understand (correct me if I am wrong), the reported results are only on examples with "high disagreement" between classifiers. However, for practical use cases, it is useful to know how the overall accuracy compares. One major risk of using the "surprisingly popular" algorithm is that the algorithm may cause the ensemble to make many incorrect predictions when the majority is right (but the minority prediction is selected). If you have those numbers, I would be interested to see them added to the paper.

I am also unsure about if applying the "surprisingly popular" algorithm in machine learning makes sense. The algorithm is motivated by the fact that difference agents have different information. However, in the ML setting, various classifiers usually have the same information. It's possible to restrict the information given to each classifier, but that would limit the performance of each individual classifier (and hurt the ensemble). I would be curious if the authors have any thoughts on this point.

I also have a few questions/concerns about how the approach is implemented:
* Why not train a single model to predict the average prediction of all models and use that model's prediction as the prior? This approach seems simpler but equivalent to the approach currently taken.
* Why not use model distillation (predicting all output logits/probabilities, or an average thereof) rather than just predicting an average of 0/1 predictions?
* For HMTS, why do the regressors for each of L labels need to be separate? It seems more efficient to use a multi-class model (as many model distillation approaches do)
* If DMTS can learn to predict when most classifiers are wrong, why wouldn't the original classifiers themselves learn to predict the answer correctly? It seems to me that the reason the experiments show that DMTS/HMTS work is that some/many of the underlying classifiers are weaker than the model that is used to ensemble the predictions (an assumption that doesn't hold in practice).


Overall, I really like the high-level idea, and a better ensembling approach promises to bring empirical gains across many ML tasks. However, I have several concerns about the experiments, motivation, and algorithmic decisions which make me hesitant to recommend the paper for acceptance.

**Experience Assessment:**

I do not know much about this area.

**Review Assessment: Checking Correctness Of Derivations And Theory:**

I assessed the sensibility of the derivations and theory.

**Review Assessment: Checking Correctness Of Experiments:**

I assessed the sensibility of the experiments.

**Review Assessment: Thoroughness In Paper Reading:**

I read the paper thoroughly.

---

> ### Author Response · Authors · 2019-11-09
> **Response to Official Blind Review #3**
>
> Thank you very much for the valuable feedback.
>
> Question 1 (“Unless I am mistaken, the authors use a more powerful…...”): HMTS only uses MLP regressors to predict the information "``how many other classifiers would agree with themselves". We use linear regressors in HMTS instead of MLPs and obtain almost the same performance. As mentioned in Section 4.2 in the paper, we tried MLP and other methods such as logistic regression in DMTS and the performance are almost the same. So using MLP regressors in HMTS or classifier in DMTS don't provide additional benefits. Other experimental results will be provided in the revision.
>
> Question 2 (“In this paper, each of the datasets we used has a small size...…”): Our paper tried 12 datasets and list their performance. Some of them are small datasets in which the training and testing datasets have only several hundreds of data points. But some of them, especially for multiclass datasets, are not small and have > 10,000 data points. Our proposed methods have improvements in both large and small size of dataset. As suggested by our experimental results, the improved performance in small size of dataset is larger.
>
> Question 3 (“As I understand (correct me if I am wrong), the reported...…”): Because the majority voting rule, or other ensemble methods, are more likely to be wrong in the "high disagreement" situation, e.g., 8 classifiers say 1 and 7 say 0, this is a high disagreement setting. We will do the experiments in all cases as you suggested and the experimental results will be provided in the revision version.
>
> Question 4 (“I am also unsure about if applying the "surprisingly popular" algorithm in machine learning makes sense....…”): We agree, if all classifiers share the same information, the idea of BTS won’t work. In practice, we will need to diversify the baseline classifiers. In our experiments, this is exactly why we add small but different amounts of noise to the training data to train different classifiers.
>
> Question 5 (“I also have a few questions/concerns about how the approach is implemented....…”):
> 5.1 (“Why not train a single model to predict the average prediction of all models....…”): Because we want to make the same setting as the seed paper. We consider each classifier as an independent agent which should have its independent model to predict its own ``"how many other classifiers would agree with themselves".
> 5.2 (“Why not use model distillation (predicting all output logits/probabilities....…”): Our proposed approaches have the probabilities firstly and then output 1 or 0 based on the comparison of probabilities.
> 5.3 (“For HMTS, why do the regressors for each of L labels need to be separate....…”): Because we considered each classifier as an independent agent.
> 5.4 (“If DMTS can learn to predict when most classifiers are wrong....…”): For different basic classifiers, there are always different types of misclassification points even though for the same type and same-level performance of strong classification models with different parameters such as deep learning methods. We want to propose approaches which can also detect whether the minority answer is correct or not in the situation mentioned above. In HMTS and DMTS, we used other methods which are similar to the basic classifiers to replace MLP models and obtained almost the same performance. So MLP models don't provide additional benefit.

---

### Official Review · AnonReviewer2 · 2019-10-24
**Official Blind Review #2**

**Rating:** 6

**Review:**

Update after author response:
I would like to thank the authors for the thoughtful response, and for addressing several of the concerns raised by the reviewers. The updated draft look cleaner and conveys the value of the proposal better. I am changing my assessment to "6: Weak Accept" (the smallest jump from 3 is to 6 in the portal. I would choose a 5 if that was possible). My concerns for the significance of results and lack of a baseline that takes worker quality into account still stand, but the results look better and the authors make a more convincing case prompting me to change my rating.
---------------------------

In this paper, the authors propose a way to measure a notion of surprise (disparity between prior and posterior) and using that as a classification rule. So, if the posterior for a class is larger than its prior, the method outputs that class label. In order to estimate the average prior for a pool of models, the method recommends building additional models to predict these “peer-priors”, whereas the posterior is simply estimated by a maximum likelihood model (P(y=1) = Indicator(y=1)/K). The authors also propose a variant that directly tries to predict when the majority answer might be wrong. The authors show that such methods can do better than majority voting and some ensemble methods on a few datasets.

The paper looks at an interesting problem of when a majority vote response might be wrong by extending the notion of surprise previously defined for human labelers to machine learning models. The strong points of the paper:
1. Simple and interpretable extension of a previously studied method.
2. The method can be plugged into existing frameworks to replace majority voting.
3. The results on the datasets considered seem good.

Here are some of my concerns:
1. The baseline of majority voting is fairly weak, since there are several models that take worker quality into account when aggregating responses. For example, the classic Dawid-Skene (1979) model. It’s not very promising to see a model just beat majority voting.
2. The presentation of results in Table 1 and 2 can be improved. Even though it’s good to know how many predictions were corrected, it will also be good to see the overall accuracy numbers (like the ones Table 3).
3. DMTS doesn’t really have the same underlying machinery as HMTS, since it doesn’t operate on the surprise measure, and putting them together in the same paper dilutes the focus.
4. What are the weights in the weighted majority?
5. The results in Table 3 show that HMTS is better than other ensemble methods but only marginally. Since HMTS uses more complex intermediate models (MLPs), I am not convinced whether the small improvement is from the proposed method or just more expressive models. For example, what would happen if the base classifiers in Adaboost were MLPs?
6. How is minority defined in multi-class scenario?
7. Minor formatting and grammatical issues: “likely to the correct”, “majority is tending to be correct”, “seemingly irrelevant topics”, “whether adopting the minority”, “aim to provide instruction to cases”, “linear regression” (should be “logistic regression” since it’s a classifier), and so on.

In summary, I think the paper has an interesting approach to an important problem, but with results that are only marginally convincing. I would have liked to see a more thorough empirical investigation to clearly establish the value of the proposed method. Based on these observations, I think the paper misses the mark and is slightly below the acceptance threshold for me.

**Experience Assessment:**

I have read many papers in this area.

**Review Assessment: Checking Correctness Of Derivations And Theory:**

I assessed the sensibility of the derivations and theory.

**Review Assessment: Checking Correctness Of Experiments:**

I carefully checked the experiments.

**Review Assessment: Thoroughness In Paper Reading:**

I read the paper thoroughly.

---

> ### Author Response · Authors · 2019-11-09
> **Response to Official Blind Review #2**
>
> Thank you very much for the useful feedback.
>
> Question 1: We agree that the simple uniformly weighted majority voting is not so strong. Therefore, we compared our methods to several popular existing ensemble methods such as weighted majority method. Experimental results show that our proposed methods obtain better performance.
>
> Question 2: The overall accuracy numbers of uniformly weighted majority, HMTS, and DMTS in 12 datasets will be listed in the revision version.
>
> Question 3: Our goal is to propose the machine learning aided methods which aim to reveal the truth when it is minority instead of majority who has the true answer. In HMTS, machine learning is used to help elicit a belief from a classifier on ``how many other classifiers would agree with themselves" which is considered as prior. Then a comparison between prior and posterior is conducted as Bayesian Truth Serum does. Overall, HMTS seems more like a heuristic algorithm. That's why we call it Heuristic Machine Truth Serum. In our opinion, another machine learning aided method is to directly train one classifier to predict whether adopting the minority as the answer or not.
>
> Question 4: We used the code (https://github.com/GRAAL-Research/majority-vote-bounds) published by the authors of <<Risk Bounds for the Majority Vote: From a PAC-Bayesian Analysis to a Learning Algorithm>> without any change. The number of their basic classifiers is 285 and the dimension of weight vector is 285 which is hard to list it here.
>
> Question 5: HMTS only uses MLP regressors to predict the information "``how many other classifiers would agree with themselves". We use linear regressors in HMTS instead of MLPs and obtain almost the same performance. As mentioned in Section 4.2 in the paper, we tried MLP and other methods such as logistic regression in DMTS and the performance are almost the same. So using MLP regressors in HMTS or classifier in DMTS don't provide additional benefits. Because we have 15 basic classifiers including Naive Bayes, Peceptron, Logistic Regression, Random Forest, and MLPs, we believe using 15 decision trees as basic estimators in Adaboost is fair. And we will try to do the related experiments as you suggested and list them in the revision version.
>
> Question 6: The class label is defined as minority if its probability is not the largest. So more than one labels can be considered as minority in multi-class scenario.
>
> Question 7: Sorry about these formatting and grammatical issues. We will correct them in our revision version.

---

### Author Response · Authors · 2019-11-13
**Revised draft uploaded. Summary of main changes; added experiment results (more datasets, comparing to stacking)**

Dear reviewers,

We have updated our draft and uploaded a revision. We would like to thank the reviewers for the helpful comments. We believe our draft has improved substantially from the last version. We would highly appreciate it if you could read our revisions and let us know if you have any further concerns.

To summarize our main changes:

1. We have improved the readability of Table 1 & 2 of our main results. Besides highlighting how many corrections our solutions are able to capture, we also added numbers of our overall accuracy.

Though these numbers seem to be incremental, we want to emphasize that this is mainly due to the relatively small number of instances the majority of classifiers got it wrong. For instance, suppose there are 20% of data points that the majority classifiers got them wrong. If our algorithms achieve consistently 10% improvement, it will be a 2% overall improvement. Our method is primarily designed to improve the robustness of ensemble methods against these “hard” or challenging instances, which are arguably the more interesting cases.

2. We have completed Table 3 with full details across all datasets. We have added comparison to stacking algorithm per the suggestion from Reviewer 1.

3. We have clarified the implication of Theorem 4.2.

4. We fixed other minor inconsistencies based on reviewers’ comments.

Best,
Authors

---

### Decision · Program_Chairs · 2019-12-19

**Decision:**

Reject

**Comment:**

This paper proposes a family of new methods, based on Bayesian Truth Serum, that are meant to build better ensembles
from a fixed set of constituent models.

Reviewers found the problem and the general research direction interesting, but none of the three of them were convinced that the proposed methods are effective in the ways that the paper claims, even after some discussion. It seems as though this paper is dealing with a problem that doesn't generally lend itself to large improvements in results, but reviewers weren't satisfied that the small observed improvements were real, and urged the authors to explore additional settings and baselines, and to offer a full significance test.